# An In Silico Functional Analysis of Non-Synonymous Single-Nucleotide Polymorphisms of Bovine *CMAH* Gene and Potential Implication in Pathogenesis

**DOI:** 10.3390/pathogens12040591

**Published:** 2023-04-13

**Authors:** Oluwamayowa Joshua Ogun, Opeyemi S. Soremekun, Georg Thaller, Doreen Becker

**Affiliations:** 1Institute of Animal Breeding and Husbandry, University of Kiel, Olshausenstraße 40, 24098 Kiel, Germany; 2The African Computational Genomics (TACG) Research Group, MRC/UVRI and LSHTM, Entebbe 5159, Uganda; 3Molecular Bio-Computation and Drug Design Laboratory, School of Health Sciences, Westville Campus, University of KwaZulu-Natal, Durban 4001, South Africa; 4Institute of Genome Biology, Research Institute for Farm Animal Biology (FBN), Wilhelm-Stahl-Allee 2, 18196 Dummerstorf, Germany

**Keywords:** sialic acid, Neu5Gc, Neu5Ac, CMAH, pathogenesis, SNPs, virus, bacteria

## Abstract

The sugar molecule N-glycolylneuraminic acid (Neu5Gc) is one of the most common sialic acids discovered in mammals. Cytidine monophospho-N-acetylneuraminic acid hydroxylase (CMAH) catalyses the conversion of N-acetylneuraminic acid (Neu5Ac) to Neu5Gc, and it is encoded by the *CMAH* gene. On the one hand, food metabolic incorporation of Neu5Gc has been linked to specific human diseases. On the other hand, Neu5Gc has been shown to be highly preferred by some pathogens linked to certain bovine diseases. We used various computational techniques to perform an in silico functional analysis of five non-synonymous single-nucleotide polymorphisms (nsSNPs) of the bovine *CMAH* (*bCMAH*) gene identified from the 1000 Bull Genomes sequence data. The c.1271C>T (P424L) nsSNP was predicted to be pathogenic based on the consensus result from different computational tools. The nsSNP was also predicted to be critical based on sequence conservation, stability, and post-translational modification site analysis. According to the molecular dynamic simulation and stability analysis, all variations promoted stability of the bCMAH protein, but mutation A210S significantly promoted CMAH stability. In conclusion, c.1271C>T (P424L) is expected to be the most harmful nsSNP among the five detected nsSNPs based on the overall studies. This research could pave the way for more research associating pathogenic nsSNPs in the *bCMAH* gene with diseases.

## 1. Introduction

Infectious diseases in cattle have an unpredictable economic impact on livestock output, and the effects of pathogens such as viruses and bacteria have been extensively investigated over the past decades. A critical feature of this research is the interaction between pathogens with host cells via sialic acids (Sias). Sias are acidic sugars with a 9-carbon backbone at the terminals of glycan chains in glycoconjugates on the surface of vertebrate cells [1]. They are components of cell surface glycans and are involved in cell communication as well as pathogen–host interactions during infectious processes. Sias are located at the terminal position and serve as the principal interface between pathogens and host cells [2].

The principal Sias in mammalian cells are N-glycolylneuraminic acid (Neu5Gc) and N-acetylneuraminic acid (Neu5Ac). Cytidine monophospho-N-acetylneuraminic acid hydroxylase (CMAH) catalyses the conversion of the Neu5Ac to the Neu5Gc molecule, which is encoded by the *CMAH* gene. In humans, Neu5Gc is absent due to the inactivation of the *CMAH* gene by mutation [2,3], and dietary metabolic incorporation of Neu5Gc has been linked to specific diseases and disorders [4].

Additionally, Sias can act as receptors for a variety of influenza viruses, including Influenza A and B [5,6,7]. Sias also regulate receptor binding by modulating transmembrane signalling, fertilisation, and cell differentiation [8]. Certain bacteria and viruses have a strong affinity for Neu5Gc, as demonstrated in different studies [9,10,11]. Schwegmann et al. [11] showed that *E. coli* K99 induces Neonatal Calf Diarrhoea (NCD) by selectively recognising Neu5Gc glycoconjugates. The findings corroborated prior research indicating that Neu5Gc glycoconjugates act as receptors [12,13]. Additionally, the bovine strain of Nebraska Calf Diarrhoea Virus, a primary pathogenic virus responsible for NCD, shows a high affinity for Neu5Gc glycoconjugates [14,15].

As sequencing technology has advanced in recent years, livestock breeding programs have reaped enormous benefits from genome sequence data [16,17]. Single-nucleotide polymorphisms (SNPs) have been utilised to investigate correlations and genetic links to segments of the genomes associated with various disorders [18,19,20]. A non-synonymous SNP (nsSNP) is an SNP that results in an amino acid substitution in the protein sequence. This mutation may impair the protein’s overall function or be linked to pathogenesis [21,22].

In silico analyses of nsSNPs in the bovine *CMAH* (*bCMAH*) gene are limited. Several research studies [23,24] have applied bioinformatics tools in determining the association of diverse nsSNPs with diseases. Additionally, studies have found SNPs in the feline and canine *CMAH* genes related to alterations in CMAH function [25,26,27]. Considering the importance of CMAH in pathogenesis [28], it is critical to find SNPs within the *CMAH* gene that may be associated with cattle diseases. Individual animals with a high level of Neu5Gc expression may be prone to certain diseases, and dietary inclusion of such cattle products may also raise the risk of certain diseases in humans [29,30]. The present study aimed to study the disease-causing nsSNP of the *CMAH* gene and evaluate its impact on the structural stability and functioning of CMAH. The study, coupled with sequencing and bioinformatics tools, facilitated the identification of pathogenic variants. The outcomes of the study facilitated an understanding of the genetic variation influence on CMAH conservation and protein stability.

## 2. Materials and Methods

### 2.1. Identification of nsSNPs in bCMAH from the 1000 Bull Genomes Sequence Data

In this study, DNA samples were analysed from a total of 165 individuals belonging to 29 different breeds and 5 unknown breeds. PCR-free fragment libraries with 300–500 bp insert sizes were prepared using the TruSeq DNA PCR-Free protocol [31] and sequenced on Illumina HiSeq3000 lanes with paired-end reads (2 × 150 bp), and the fastq files were created using Casava 1.8. The paired-end reads were then mapped to the cow reference genome UMD3.1/bosTau6 and aligned using the Burrows–Wheeler Aligner (BWA) version 0.5.9-r16, with default settings [32]. The SAM file generated by the BWA was then converted to BAM, and the reads were sorted by chromosome using samtools (http://samtools.sourceforge.net, accessed on 5 April 2023). The PCR duplicates were marked using Picard tools (http://sourceforge.net/projects/picard/, accessed on 5 April 2023). The Genome Analysis Tool Kit (GATK version 2.4.9 [33] was used to carry out local realignment and to produce a cleaned BAM file. Variant calls were then made with the unified genotype module of GATK. The variant data for each sample were obtained in variant call format (.vcf), as were raw calls for all samples and sites flagged using the variant filtration module of GATK. Variant filtration was carried out, following the best practice documentation of GATK version 4. The snpEFF software [34], together with the UMD3.1/bosTau Ensembl annotation, was used to predict the functional effects of the detected variants. Based on the obtained information, DNA sequence data from 2724 individuals were retrieved and analysed from the 1000 Bull Genomes sequence data [35]. For the sequence analysis and identification of different nsSNPs, the exon table of the mRNA transcript variant X6 (XM_024984024.1), comprising 16 exons, of which 14 are coding, was used from the NCBI database (https://www.ncbi.nlm.nih.gov, accessed on 5 April 2023). The total spliced RNA is 3374 bp long and encodes for the protein isoform X2 (XP_024839792.1), which is 577 amino acids (AA) in length. The nsSNPs were retrieved between the genomic positions 32,458,973 bp and 32,755,484 bp of the bovine chromosome 23 (NC_037350.1).

### 2.2. Prediction, Refinement, and Validation of Tertiary Structure of Bovine CMAH Protein

Due to the absence of the tertiary structure for the bCMAH protein in the protein database and sequence homology between the target and template proteins of less than 30%, we used ab initio structure prediction with all-atom refinement via the Robetta online tool (https://robetta.bakerlab.org/, accessed on 5 April 2023) [36]. The TrRosetta (TR, a method based on deep learning) was utilised as the default. A Monte Carlo minimisation methodology was employed that involves perturbing randomly chosen backbone torsion angles while optimising sidechain rotamer conformations and performing Quasi-Newton minimisation on all backbone and sidechain torsion angles [36]. Protein modelling was performed using the sequence of the bCMAH protein isoform X2 (XP_024839792.1). Further refining of the protein structure was carried out using the GalaxyWEB refiner tool (https://galaxy.seoklab.org/index.html, accessed on 5 April 2023). The server refines loop or terminal areas using ab initio modelling and then uses molecular dynamics simulations to execute repetitive structure perturbation and eventual overall structural relaxation [37]. The predicted bCMAH protein structure was further validated using the PDBsum (http://www.ebi.ac.uk/thornton-srv/databases/pdbsum/Generate.html, accessed on 5 April 2023) [38] and ProSA online tools (https://prosa.services.came.sbg.ac.at/prosa.php, accessed on 5 April 2023) [39]. The domain information of CMAH in bovine was obtained through the UniProt database (https://www.uniprot.org/, accessed on 5 April 2023 [40]) and PROSITE (https://prosite.expasy.org/, accessed on 5 April 2023 [41]). The disordered region on the protein was predicted through the D2P2 webserver tool [42].

### 2.3. Evaluation of the Functional Impacts of the nsSNPs on the Function and Stability of the Bovine CMAH Protein

To assess the functional effects of nsSNPs and to deduce their potential role in pathogenesis, we used a combination of five different computational online tools based on different algorithms. The primary amino acid sequence of bCMAH was uploaded into various tools. The online computational tools were: PolyPhen-2 (http://genetics.bwh.harvard.edu/pph2/, accessed on 5 April 2023) [43], SNPs&GO (https://snps.biofold.org/snps-and-go/, accessed on 5 April 2023) [44], Sorting Intolerant From Tolerant (SIFT; https://sift.bii.a-star.edu.sg/www/SIFT_dbSNP.html, accessed on 5 April 2023) [45], Protein Variation Effect Analyzer (PROVEAN; http://provean.jcvi.org/index.php, accessed on 5 April 2023) [21], and PANTHER (http://www.pantherdb.org/, accessed on 5 April 2023) [46]. The overall criteria for the classification of nsSNP into deleterious and benign classes were taken from [47].

The influence of variations was further evaluated for their effect on the structural stability of the bCMAH protein. The 3D structure of bCMAH was uploaded into DynaMut (http://biosig.unimelb.edu.au/dynamut/, accessed on 5 April 2023), an online computational tool, and SNP information was fed into it [48]. DynaMut investigates the effects of point mutations on the dynamics and stability of proteins due to variations in vibrational entropy. Additionally, it incorporates graph-based signatures and normal-mode dynamics to produce a consensus forecast [48].

### 2.4. Sequence Conservational Analysis

The web-based program ConSurf (http://consurf.tau.ac.il, accessed on 5 April 2023) was used to analyse sequence conservation. ConSurf analyses the evolutionary trend of functional region amino acids. It classifies the protein’s amino acid residues on a scale of 1 to 9, with 1–3 denoting variable, 4–6 denoting average, and 7–9 denoting conserved or highly conserved sections [49]. Variations that fell into the conserved region were taken as the most deleterious.

### 2.5. Post-Translational Modification Sites’ Prediction

In order to fully comprehend protein activities and regulation, it is critical to identify and analyse post-translational modification sites (PTMs). By inputting the protein FASTA sequence of the bCMAH, the PTMs online tool MusiteDeep (https://www.musite.net, accessed on 5 April 2023), a deep learning framework, was utilised to predict the PTMs [50]. The tool incorporates numerous ensemble tools that facilitate better prediction of PTM sites.

### 2.6. Stability Analysis

The predicted tertiary structure of CMAH was analysed using I-Mutant v2 to evaluate the impact of nsSNPs on the structural stability (https://folding.biofold.org/i-mutant/i-mutant2.0.html, accessed on 5 April 2023 [51]). Any nsSNP with a ΔΔG value lower than −0.5 was considered to have a destabilizing effect.

### 2.7. Active Site Prediction and Molecular Dynamic Simulation

To identify potential binding sites within the tertiary structure of the bCMAH protein, we used SiteMap 3.5, inbuilt within the Schrödinger software package [48], to characterise those binding sites. The SiteMap provides quantitative and graphical information that can help guide efforts to critically assess virtual hits in a lead-discovery application or to modify the ligand structure to enhance the potency or improve physical properties in a lead-optimisation context [52]. Further validation of the predicted active sites was carried out using MetaPocket 2.0 [53].

In this study, the PyMol v4.0.4 mutagenesis wizard tool was employed to introduce point mutations in the predicted CMAH structure [54]. To evaluate the stability of the predicted and mutagenic structures, molecular dynamics (MD) simulations were carried out using the CHARMMS forcefield in GROMACS 2016 [55]. The system was prepared by first subjecting it to solvation, followed by the addition of SPC216 water molecules, and then neutralization with Na+/Cl− ions. An energy minimisation (EM) step was performed using the steepest descent method with a total timestep of 50,000 steps to obtain an optimised system. The modelled system was then subjected to NVT (constant number of particles, volume, and temperature) and NPT (constant number of particles, pressure, and temperature) equilibration for 100 ps. Trajectories were initiated from the same random seed to minimize any biases during the MD simulations.

Following equilibration, MD simulations were carried out for 50 ns, and trajectory coordinates were captured every 10 ps. Structural analysis was performed using GROMACS 2016 built-in programs. Trajectories were constructed using gmx_trjconv, and the root mean square deviations (RMSD) were calculated using gmx_rms. The root mean square fluctuations (RMSF) were computed using gmx_rmsf, the radius of gyration (Rg) using gmx_gyrate, the number of hydrogen bonds using gmx_bond, and the solvent accessibility surface area (SASA) using gmx_sasa. RMSD, RMSF, and Rg calculations were carried out for the backbone. The data were represented through scatter smooth-line plots to visualize any changes in the protein structure. These analyses helped to determine the stability of the predicted and mutagenic structures. Overall, the study aimed to provide a deeper understanding of the effects of the introduced point mutations on the CMAH structure and to identify any potential changes in its stability.

## 3. Results

### 3.1. Distribution of nsSNPs in Different Breeds Analysed in 1000 Bull Genomes Sequence Data

The new genomic positions and coding exon variants of the bovine *CMAH* gene sequenced in 165 individuals were identified in the UMD_3.1.1 and re-mapped on ARS-UCD1.2 assemblies, resulting in the identification of novel variants, as well as the confirmation of previously known variants through their corresponding RefSNP I.D. A total of 2724 DNA sequences were examined and classified into 3 categories: 13 dairy breeds (1349 samples), 9 beef breeds (774 samples), and 9 dual-purpose or crossbred types (601 samples). Within the *bCMAH* gene, five non-synonymous single-nucleotide polymorphisms (nsSNPs) were discovered, which are predicted to result in missense mutations (Table 1). The frequencies of the identified genotypes in the 1000 Bull Genomes sequence dataset are displayed in Table 2. The nsSNP c.319A>G, located in exon 4, exhibited the highest frequency in both heterozygous and homozygous forms. Conversely, the nsSNP c.1271C>T demonstrated the lowest frequency, appearing solely in a heterozygous state.

### 3.2. Secondary and Tertiary Structure Prediction and Validation

To validate the tertiary structure, the PDBsum’s PROCHECK tool was applied, and the results provided a thorough breakdown of the protein’s composition. According to PDBsum, the protein includes a strand containing 89 residues (15.4%), an alpha helix with 129 residues (22.4%), a helix with 13 residues (2.3%), and other components with 346 residues (60.0%). Additionally, K107 and N242 are predicted to be strand turns, while A210S, P424, and F512 are anticipated to be helix turns (Figure 1). The Ramachandran plot analysis, which assesses the stereochemical quality of the tertiary structure, revealed that an impressive 92.6% of the residues were in the most favoured regions. ProSa, an interactive web service designed for identifying errors in tertiary protein structures [39], demonstrated that the modelled protein achieved a z-score of −7.78. This score is indicative of a high-quality structure. Furthermore, the protein structure was found to be consistent with the standard X-ray crystallography parameters for proteins of a similar size (Figure 2). We also predicted a disordered region in the bCMAH protein that showed a high presence of disorder and provided insight into the presence of loops in the CMAH protein’s structure.

### 3.3. Prediction of Pathogenic and Damaging Amino Acids of Bovine CMAH Protein

Identifying variants associated with pathogenesis is critical, as this might help determine the druggability of such variations. The matching results provided by the five tools (Polyphen-2, SNPs&Go, PROVEAN, SIFT, and PANTHER) independently demonstrated the reliability of the predictions, despite using different algorithms. The K107E, A210S, N242S, and F512Y variants were expected to be neutral, tolerable, or benign, while the P424L variant was anticipated to be probably harmful, deleterious, not tolerated, or disease-causing (Table 3).

### 3.4. Prediction of the Effects of Amino Acid Substitutions on Bovine CMAH Protein Stability

In this study, we used the DynaMut online program to investigate the structural implications of amino acid substitutions in a protein of interest. The program utilizes a computational approach to predict the impact of mutations on protein stability. To carry out this analysis, we uploaded the tertiary structure of our protein in PDB format to the DynaMut webserver. Our analysis revealed that the substitutions K107E, A210S, and N242S (Figure 3A–C) were predicted to stabilize the protein structure. On the other hand, the substitutions P424L and F512Y (Figure 3D,E) were projected to destabilize the protein. These findings are consistent with previous research showing that mutations can have both stabilizing and destabilizing effects on protein structures. Mutations also altered molecular interactions within the protein. Mutation K107E led to the loss of the hydrogen bond, mutation A210S led to the loss of hydrogen bonds as well as hydrophobic interactions, mutation N242S introduced hydrogen bonds, mutation P424L caused the loss of aromatic contact, and mutation F512Y caused the loss of hydrogen bonds.

To gain a deeper understanding of the interatomic variations between the variants and wildtype residues, we examined the associated changes in vibrational entropy. Our analysis revealed that the substitutions K107E, A210S, and N242S resulted in decreased vibrational entropy, indicating a more ordered protein structure. In contrast, the substitutions P424L and F512Y led to increased vibrational entropy, indicating a more disordered protein structure [43].

### 3.5. Sequence Conservational Analysis and the Predicted PTMs

Conserved regions of protein sequences and PTMs are vital in analysing disease-causing or structure-altering mutations [56,57]. The FASTA protein sequence was analysed using the ConSurf web tool. ConSurf predicted K107E to be present in a variable region, A210S and N242S in the average region, and the last two variants (P424L and F512Y) in the highly conserved region (Figure 4). Two variants were also predicted to be located in the PTMs by the MusiteDeep webserver tool. While variant K107E was predicted to be associated with ubiquitination, SUMOylation, acetylation, methylation, or hydroxylation, the variant P424L was only associated with hydroxylation (Table 4).

### 3.6. Active Sites’ Prediction and Molecular Dynamics Simulations

The active site of bCMAH was predicted using Sitemap3.5 [49] and cross-validated with MetaPocket2.0 [53]. This binding pocket was chosen based on the druggability (Dscore) score, which measures a protein’s ability to bind small molecules tightly. Based on the Dscore, tentative active site residues and three locations were predicted, as shown in Table 5 and Figure 5. The results show that none of the amino acid substitutions were located in the predicted active sites.

#### 3.6.1. Bovine CMAH Mutations’ Impact on Its Structural Stability

The impact of bCMAH mutations on its structural stability was also evaluated through I-Mutant. The analysis revealed that all mutations promoted structural stability in bCMAH. The stability impact was predicted through DDG values that indicate the delta-free energy. Table 6 indicates each mutation’s impact on the structure and function of the bCMAH protein.

#### 3.6.2. Mutations Elicited Structural Distortion in Bovine CMAH Protein

To understand the structural perturbation of bovine CMAH caused by the polymorphisms, we used root mean square deviation (RMSD), the radius of gyration (Rg) and root mean square fluctuation (RMSF) to characterise the structural events in the proteins during the 50 ns simulation. Compared to the wildtype structure, an increase in the RMSD of the mutant F512Y was observed. Mutants K107E, A210S, N242S, and P424L RMSD values were lower than the wildtype (Figure 6A). The wildtype’s Rg decreased after 20 ns, suggesting that the wildtype structure compacts and reaches the point where its activity diminishes. However, all mutations enhanced the Rg of bCMAH, and the values remained in a closer range throughout the simulation (Figure 6B).

RMSF indicates the influence of mutations on the stability of the domain in which they are located and the nearby regions. High fluctuations in residues 21–39 were observed in the wildtype, and all the mutant structures had a lower fluctuation in these residues. However, mutant K107E had lower fluctuations than the wildtype from residues 92–109. Mutant N242S showed high fluctuations in residues 421–520 and had overall lower fluctuations at its C-terminus (Figure 6C). Among the mutant structures, mutant A210S was found to be the most stable. No significant impact of mutations on the number of hydrogen bonds and solvent accessible surface areas (SASA) was recorded (Figure 6D,E). Overall, compared to the wildtype, all mutations promoted the structural stability of the bCMAH protein, whereas the mutation A210S greatly enhanced bCMAH stability, while the stability of the F512Y mutant was comparatively lower than the other mutants.

## 4. Discussion

Neu5Gc and Neu5Ac are mammals’ predominant sialic acid sugar molecules. While both Neu5Gc and Neu5Ac are prevalent in bovine species, the Neu5Gc cannot be endogenously synthesised in humans because the CMAH is inactivated [2,3]. On the one hand, the association between Neu5Gc and some bovine diseases is due to the affinity of specific pathogens for Neu5Gc glycoconjugates [14,15]. On the other hand, dietary incorporation of Neu5Gc sugar molecules via red meat has been associated with various human diseases and disorders [4]. As previously noted, cattle expressing a high level of Neu5Gc may be prone to certain diseases, and intake of their meat products may enhance human vulnerability to certain diseases.

SNPs are frequent genetic variants that occur approximately every 500–1000 base pairs and are useful for genome association and pharmaco-genomic investigations [58]. Computational or in silico analysis is becoming more and more popular for mapping SNPs to changes in protein functions or diseases [24,47,59]. Apart from being cost- and time-effective, other studies have demonstrated the efficacy of various computational or in silico analysis tools in precisely identifying SNPs associated with various diseases or changes in protein functions [23,24]. The nsSNPs that lead to changes in the amino acid sequence of a protein may disrupt the protein’s overall tertiary structure, which can result in diseases and disorders [21,45].

This study identified five non-synonymous single-nucleotide polymorphisms (nsSNPs) within the *bCMAH* gene using data from the 1000 Bull Genomes project [35]. This dataset comprises whole-genome sequences from a diverse range of cattle populations across the globe. The majority of nsSNPs were observed in heterozygous and homozygous states. The c.319A>G variant displayed the highest frequency for both heterozygous and homozygous genotypes, while the c.1271C>T variant showed the lowest frequency as a heterozygous genotype. Previous research has linked CMAH variations to blood subtypes in various cat species [26,60].

Multiple computational tools were employed to assess the potential impact of these nsSNPs, including PolyPhen-2, SNPs&GO, PROVEAN, SIFT, and PANTHER. Despite utilizing different algorithms, all tools reached a consensus based on the provided amino acid sequence. This consensus allowed for a more reliable determination of the potential consequences of these nsSNPs. The c.1271C>T variant was anticipated to be associated with a disease or disorder, harmful, probably detrimental, or intolerable. This could account for its low frequency in the 1000 Bull Genomes sequence data.

For thoroughly evaluating the impact of these variations on the bCMAH protein structure, its structure was predicted using an ab initio approach. The obtained structure was found to be high in disorder. Intrinsically disordered proteins (IDPs) play essential roles in various cellular regulatory processes. Their lack of a well-defined structure in their free state, and sometimes when interacting with physiological partners, is a fundamental aspect of their functionality. Disordered domains often contain numerous sites for post-translational modifications, which serve as crucial elements in cellular metabolic control. Upon binding to a partner, a disordered domain may fold, leading to the formation of a complex that buries a substantially larger surface area compared to interactions between similarly sized folded proteins. This characteristic allows IDPs to maximize specificity while maintaining a relatively small size [61,62]. However, further analysis investigating the functional role of the disorder in CMAH protein interactions will be useful.

Proteins are dynamic in nature, and the effects of amino acid substitutions on their dynamics and stability may affect the protein’s function and may be associated with disease [45,48]. The DynaMut online tool projected the effects of these five mutations on the conformation, flexibility, and stability of proteins due to variations in vibrational entropy. K107E, A210S, and N242S variants were anticipated to stabilise the protein shape, whereas P424L and F512Y variants were predicted to disrupt the protein’s stability.

Additionally, the degree of conservation of the amino acid sequence of proteins strongly correlates with the functional sections of proteins, such as motifs [49]. Generally, variants in highly conserved regions are not tolerated, and those discovered within these areas may affect the protein’s function and contribute to disease [56,57]. According to the ConSurf online tool for analysing evolutionarily conserved regions, both P424L and F512Y are projected to be located in a highly conserved region, K107E in the variable region, and A210S and N242S in the averagely conserved region.

Neu5Gc’s absence or presence in felines is used to identify blood groups. Its absence is linked with type B, which might be caused by variations in the CMAH’s 5′UTR region [25,63]. Furthermore, CMAH variations were reported to have a deleterious influence in cats. Kehl et al. further found the association of the *CMAH* gene’s deleterious variations with blood group types. They reported that the c.179G>T variant in a Turkish cat breed is linked with blood type B and is reported to be deleterious [60]. Uno et al. identified 15 SNPs (11 intronic and 4 exonic) in 11 dog breeds. The nsSNP, c.554 A>G, is reported to be majorly distributed among canines. However, no loss of function or gain of function mutations with severe consequences is reported in canines [27].

PTMs are also crucial protein locations because they promote proteome diversity, which is required for biological processes such as protein–protein interactions and disease-related signalling cascades. Variants in these locations might be associated with disease [64]. Only K107E and P424L variants were predicted to be related to PTMs using the MusiteDeep online tool. While K107E was anticipated to be linked with a variety of PTMs, including ubiquitination, SUMOylation, acetylation, methylation, and hydroxylation, P424L was projected to be exclusively involved in hydroxylation. Proline hydroxylation is critical for protein stability; for example, it aids in the effective twisting of the collagen helix [65]. Proline hydroxylation is also required to control hypoxia-induced factor-1 alpha (HIF-1), a critical oxygen-dependent transcription factor [66]. Any variant detected in these PTMs may be harmful, implying that variants in the proline hydroxylation region of the bovine CMAH protein may also be associated with diseases or changes in the protein’s function.

Additionally, MD simulations were applied to evaluate the dynamics of the protein following mutation by examining the protein movements and tracking the structural changes of wildtype and mutant proteins over time using GROMACS. The 50 ns MD simulation revealed that all variations enhanced the overall stability of the CMAH protein. As the CMAH enzyme is responsible for Neu5Ac’s catalytic conversion to Neu5Gc, the enhanced stability of CMAH due to mutations might enhance the reaction rate of Neu5Gc production. Such mutations might lead to an increase in the biosynthesis of Neu5Gc, which can also be lethal for humans, causing diseases such as atherosclerosis and cancer, as earlier mentioned. Further investigation describing CMAH mutations’ impact on its interaction with Neu5Ac will further facilitate understanding of the pathogenicity of CMAH in diseases.

## 5. Conclusions and Recommendations

In silico analysis is a cost-effective and time-efficient method for analysing nsSNPs related to protein structural changes and pathogenicity. The functional and structural consequences of nsSNPs in the bovine *CMAH* gene were studied utilising a number of computational prediction tools. Five nsSNPs were identified, with c.1271C>T (P424L) having the lowest frequency in the 1000 Bull Genomes sequence data and being expected to be pathogenic or intolerable. Additionally, this P424L variant, similar to F512Y, was predicted to be present in the highly conserved region and destabilise the protein structure due to changes in vibrational entropy. It was predicted that P424L and K107E variants are located in PTMs. Consensus results from all computational techniques indicate that P424L variation may be relevant for confirming the structural disruption of the bovine CMAH protein and its association with pathogenesis. Although the MD simulation revealed that mutation A210S might significantly enhance the CMAH protein stability, based on consensus information, the P424L might be the most harmful mutation. The fundamental limitation of this study is that most computational methods utilised were optimised for the human genome. However, similar computational techniques have been successfully used for in silico analysis of SNPs in non-human species [67,68,69]. In vitro analysis, for example, site-directed mutagenesis, is recommended for future studies to validate the impacts of these nsSNPs, particularly for c.1271C>T (P424L).

## Figures and Tables

**Figure 1 pathogens-12-00591-f001:**
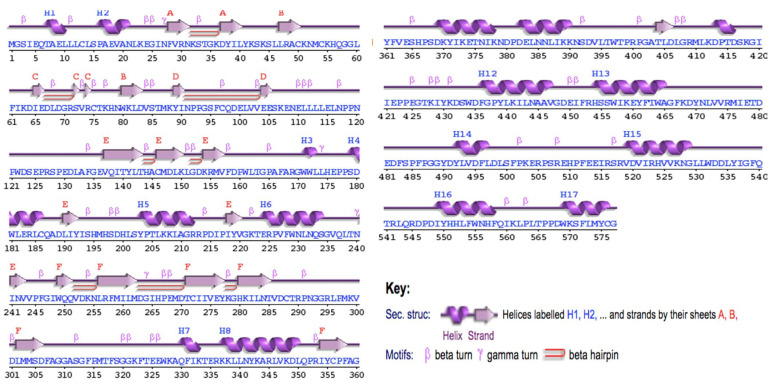
Secondary structure of bovine CMAH protein.

**Figure 2 pathogens-12-00591-f002:**
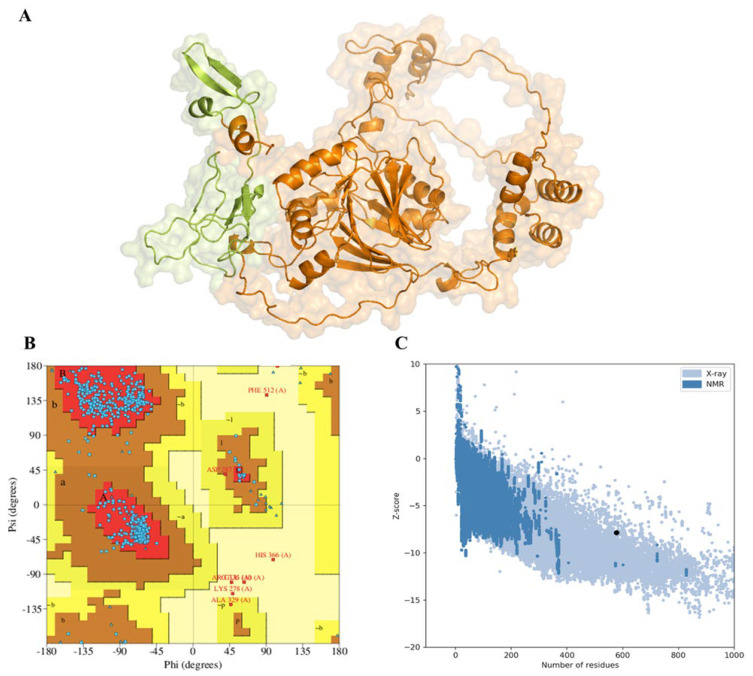
The predicted tertiary structure of bovine CMAH protein and its validation. (**A**) Tertiary structure of CMAH. Lime colour indicates iron-sulphur domain (14–112). (**B**) Assessment and validation of the protein. Ramachandran plot depicted that 92.6% of the amino acids are located in the most favoured region with a total of 462 residues (A, B, L), 5.8% in the additional allowed region with a total of 29 residues (a, b, l, p), 0.4% in the generously allowed region with two residues (~a, ~b, ~l, ~p), and 1.2% in the disallowed region with six residues. (**C**) Prosa plot with a z-score of −7.86. The black dot represents the position of the bCMAH structure compared with the standard X-ray crystallography parameters for proteins of a similar size.

**Figure 3 pathogens-12-00591-f003:**
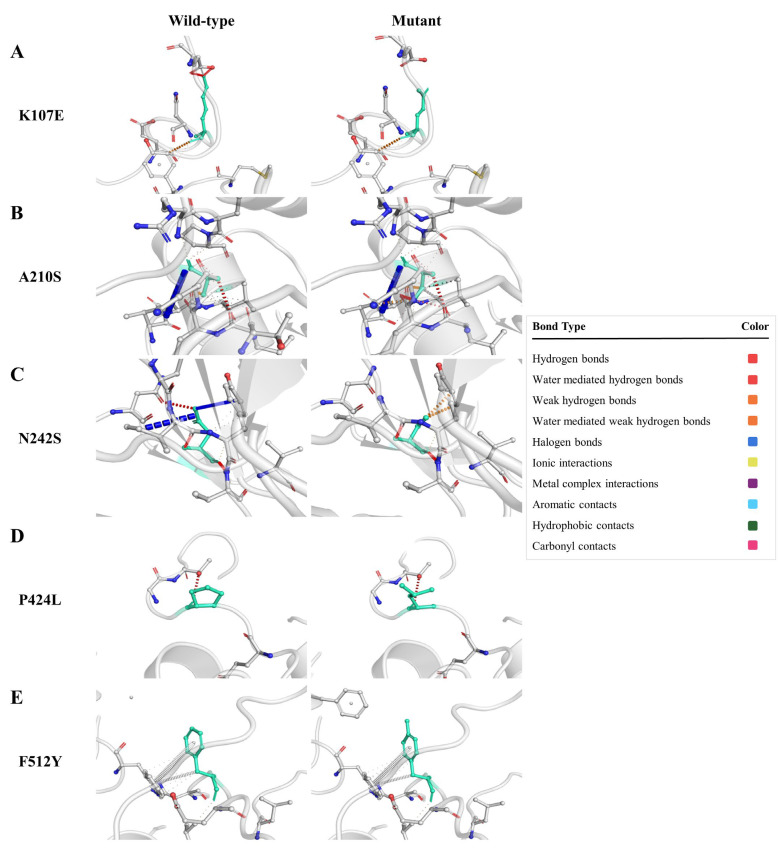
Structural impacts of the amino acid substitutions on protein stability, computed by DynaMut. K107E variant (**A**), A210S variant (**B**), N242S variant (**C**), P424L variant (**D**), and F512Y variant (**E**), with prediction outcome, ΔΔG, of 0.496, 0.271, 0.118, −0.121, and −0.804 (kcal/mol), respectively. ΔΔG = vibrational entropy change.

**Figure 4 pathogens-12-00591-f004:**
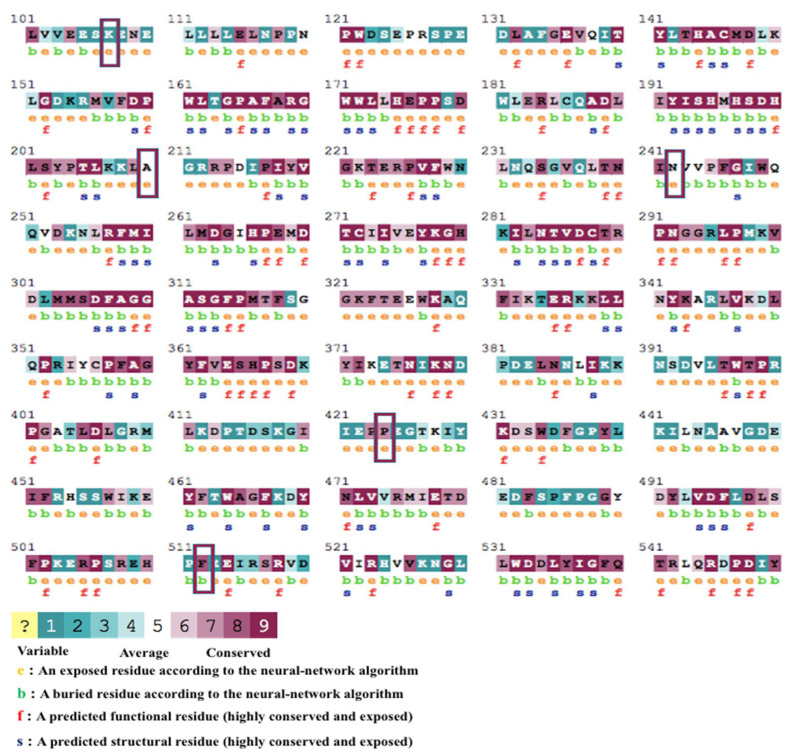
ConSurf results for residue conservation. The colours show different confidence levels for sequence conservation, with dark green being highly variable and dark red being highly conserved. The five rectangular boxes depict the conversation confidence levels of the five variants (Table 4).

**Figure 5 pathogens-12-00591-f005:**
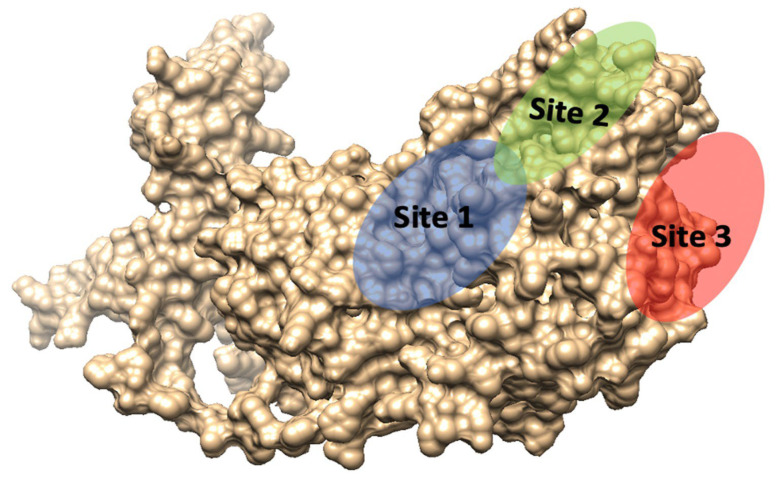
Potential active sites predicted by SiteMap and cross-validated by MetaPocket. Site 1 (blue), Site 2 (green) and Site 3 (red).

**Figure 6 pathogens-12-00591-f006:**
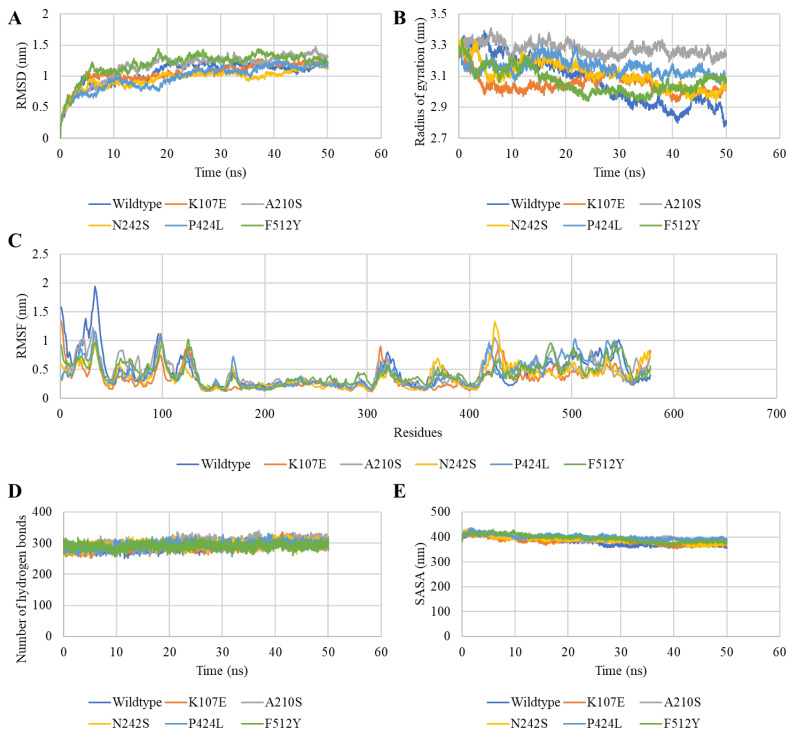
Plots depicting the MD simulation analysis of the impacts of mutations on bCMAH. Wildtype (black), K107E (red), P424L (green), A210S (blue), N242S (lemon), and F512Y (pink) across the MD simulation run. RMSD (**A**). Rg (**B**). RMSF (**C**). Hydrogen bonds analysis (**D**). SASA analysis (**E**).

**Table 1 pathogens-12-00591-t001:** nsSNPs detected within bovine *CMAH* in the 1000 Bull Genomes dataset compared with the reference sequence.

Coding Exon	cDNA Variant (XM_024984024.1)	Protein Variant (XP_024839792.1)	Genomic Position	RefSNP ID (dbSNP)
4	c.319A>G	K107E	BTA23:g.32,721,570	rs208635220
6	c.628G>T	A210S	BTA23:g.32,727,639	rs435799892
6	c.725A>G	N242S	BTA23:g.32,727,736	rs109811989
11	c.1271C>T	P424L	BTA23:g.32,743,918	rs518400910
12	c.1535T>A	F512Y	BTA23:g.32,745,866	rs380571713

Reference: ARS-UCD1.2.

**Table 2 pathogens-12-00591-t002:** The frequencies of the bovine CMAH genotypes identified.

Coding Exon	cDNA Variant (XM_024984024.1)	Heterozygous (n)	Homozygous Alternative Allele (n)	Reference Allele (n)	Null Data (n)	Total (n)
4	c.319A>G	1091	362	1140	131	2724
6	c.628G>T	40	4	2662	18	2724
6	c.725A>G	234	27	2443	20	2724
11	c.1271C>T	15	0	2689	20	2724
12	c.1535T>A	77	14	2610	23	2724

Null data = no sequence data at the position, n = number of samples.

**Table 3 pathogens-12-00591-t003:** Protein variants analysed by PolyPhen-2, SNPs&GO, PROVEAN, SIFT, and PANTHER.

Protein Variant (XP_024839792.1)	PolyPhen-2	SNPs&GO	PROVEAN	SIFT	PANTHER
K107E	Benign	Neutral	Neutral	Tolerated	Probably Benign
A210S	Benign	Neutral	Neutral	Tolerated	Possibly Damaging
N242S	Benign	Neutral	Neutral	Tolerated	Probably Benign
P424L	Probably Damaging	Disease	Deleterious	Deleterious	Probably Damaging
F512Y	Benign	Neutral	Neutral	Tolerated	Probably Benign

Predictions of the impacts of amino acid substitutions as a result of nsSNPs on bovine CMAH using different computational tools.

**Table 4 pathogens-12-00591-t004:** Conservational analysis of the variants and the predicted PTMs.

Protein Variant (XP_024839792.1)	Conservation	PTMs
K107E	Variable	Ub, Su, Ac, Me, Hy
A210S	Average	None
N242S	Average	None
P424L	Highly Conserved	Hy
F512Y	Highly Conserved	None

ConSurf conservational scale: Using the UniRef90 protein database. PTMs by MusiteDeep with a PTM threshold score of 0.05: Ub (Ubiquitination), Su (SUMOylation), Ac (Acetylation), Me (Methylation), Hy (Hydroxylation).

**Table 5 pathogens-12-00591-t005:** Highlights of the residues making up the active sites, with a description of the physicochemical properties of each active site.

Site	Site Score	Size	Volume	DScore	Residues
1	1.017	90	184.53	0.961	Ser72, Cys75, Thr76, Asn79, Asp83, Val84, Ser85, Thr86, Met87, Lys88, Pro93, Gly94, Ser95, Phe96, Lys222, Met262, Asp263, Gly264, Ile265, His266, Pro267, Glu268, Asp270
2	1.016	214	619.11	1.057	Tyr41, Lys42, Ser43, Leu46, Arg48, Lys51, Cys54, Lys55, Leu60, Thr163, Gly164, Pro165, Ala166, Phe167, Ala168, Gly170, Trp171, Trp172, Leu173, Leu174, His175, Pro177, Pro178, Trp181, Met196, His197, Ser198, Leu201, Ser202, Tyr203, Pro204, Lys208, Pro226, Val227, Trp229, Asn230, Leu231, Asn232, Gln233, Glu513, Glu514
3	0.962	71	154.69	1.001	Pro381, Asp382, Leu384, Asn385, Val394, Thr396, Trp397, Thr398, Lys468, Asp469, Leu530, Leu531, Leu535

**Table 6 pathogens-12-00591-t006:** Impact of mutations on bovine CMAH structural stability.

Mutation	DDG	Stability
K107E	−0.17	Increase
A210S	−0.44	Increase
N242S	−0.49	Increase
P424L	−0.32	Increase
F512Y	−0.33	Increase

## Data Availability

Not applicable.

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
