# Peer review of "An In Silico Functional Analysis of Non-Synonymous Single-Nucleotide Polymorphisms of Bovine CMAH Gene and Potential Implication in Pathogenesis"

_pathogens, 2023, doi:10.3390/pathogens12040591_

Round 1

Reviewer 1 Report

The article submitted Ogun et al to Pathogens explored the in silico functional analysis of the nsSNPs of bovine CMAH through various computational technique and identified P424L as one of the most harmful nsSNP that could be linked to the bovine diseases.

The article is very well written with detailed methodology which could be applied to other studies in exploring the SNP polymorphism to identify the genetic links to segments of genomes associated with the various diseases or disorders. I have very few queries which I’m listing below.

(i) It would be great if authors can mention if any literature is there which shows that the site similar to the location of P424L of bovine CMAH in feline or canine CMAH is related to any deleterious effect.

(ii) The c.1271C>T is found only in 15 heterozygous state from 2689 in total reference alleles. Is there anything specific for those 15 samples?

(iii) It is mentioned P424L and F512Y were projected to destabilize the protein structure. Will there be any interacting moieties of any other protein that is going to interact with those sites and is going to be affected due to the change in hydroxylation and overall protein structure. I mean will the mutation affects the interaction to other proteins as well.

Author Response

(i) It would be great if authors can mention if any literature is there which shows that the site similar to the location of P424L of bovine CMAH in feline or canine CMAH is related to any deleterious effect.

Response: Authors thank the reviewer for the suggestion. We found no information related to the variation P424L in feline or canine. However, we found information related to other CMAH variations, including 5'UTR, which we have incorporated in the discussion section (page 17, Line 521-525).

(ii) The c.1271C>T is found only in 15 heterozygous state from 2689 in total reference alleles. Is there anything specific for those 15 samples?

Response: The c.1271C>T variant was identified in 15 out of 2689 total reference alleles, only in the heterozygous state. We did not observe anything specific in those 15 samples. However, such an occurrence either can result from random occurrence, given the natural variation in the genome or could still affect the phenotype of the individuals carrying the variant.

(iii) It is mentioned P424L and F512Y were projected to destabilize the protein structure. Will there be any interacting moieties of any other protein that is going to interact with those sites and is going to be affected due to the change in hydroxylation and overall protein structure. I mean will the mutation affects the interaction to other proteins as well.

Response: Our research was focused on determining how these mutations affect the CMAH protein's structure and stability. By bioinformatics tools, we have shown that the P424L and F512Y mutations do result in protein structural instability. Because of potential alterations in hydroxylation and protein structure, particularly P424L may affect protein-protein interactions. Alterations to the network of protein-protein interactions may have functional repercussions. However, the literature survey indicated the unavailability of significant information related to the CMAH domains and their function as well as variations impact on CMAH protein-protein interactions. As the hypothesis of the present study was to investigate the impact of these variations on CMAH protein structural stability, the variation impact on its protein-protein interaction will be part of the future project. The reviewer highlighted an important aspect worth exploring to better understand the CMAH functioning.

Reviewer 2 Report

The study although based on in silico analysis is important and the article is well presented.

The article investigated the pathogenic potential of nsSNPs in CMAH gene that may be associated with human diseases.

The work appears original and has relevance in management/treatment of disease. The authors have used several tools and techniques to arrive at their conclusion. Comprehensive analysis led to identification of the critical SNP.

The article is well written and results and discussions well presented.

In future I would suggest the authors validate their conclusion with in vitro studies.

Author Response

We appreciate you taking the time to reviwe the manuscript, and we plan to include the in vitro mutagenesis study in future work.

Reviewer 3 Report

This study aimed to discover potentially disease-causing nsSNPs in bovine CMAH gene using sequence data from the 1,000 Bull Genomes sequence data. The authors modeled the three-dimensional structure of the CMAH protein and used computational tools to achieve the objetctive. However, although interesting, the computational analysis performed has serious flaws, and because of this, the results presented should not be used alone for establishing disease-genotype relations without any experimental validation, mostly based on ab initio algorithms for protein modelling.

Lines 62-64: This is a very delicate point, and the authors should be caraful in order to state this. Rafaee et al. [26] for instance, stated that "However, the in silico based nsSNP predictions are not solely adequate for deriving genotype–phenotype relations and more experimental investigation is needed to elucidate the impact of the I661T changes on the AKAP3 structure in these patients"

Lines 67-71: On what basis are the authors raising this hypothesis? I am not sure if this assumption makes any sense.

Line 75: In general, the methodological procedures need to be better described, not just describe what each program does.

Lines 99-102: The authors should specify what was examined in these tools. What kind of analyses were performed? What parameters were used?

Line 103: Due to the uncertain nature of the molecular modeling approaches used by the authors, it is critical that molecular dynamics simulations be performed to assess the stability of the predicted model and therefore determine which samples in the simulation are stable when compered to the initial model to be used in subsequent analyses. There is insufficient theoretical evidence that the protein model used in these analyzes is reliable.

Lines 181-183: Is this categorization important? There is no mention of that in Introduction, Objetctive and Materials and Methods!?!

Line 184: How were the nsSNPs identified? This was not described in Materials and Methods!

Lines 194-201: This is not Results!

Line 204: 60% of the protein was characterized as "others"! The authors should better characterize the structure of the protein!

Figure 2: Although the assessment parameters used indicate satisfactory numbers, the protein 3D structure presents several long loops and turns arranged in an unconventionally conformation. Some CMAH proteins of other organisms have been modelled and it is clear the difference, as one can see in PDB. Therefore, I am not sure if this model really ressembles the native protein.

Lines 273-282: The RMSD values are extraordinarily high. This indicates at least significant conformational changes if not at least partial unfolding, even for the "wild-type" model. In addition, the RMSD figure shows that 20 ns was not sufficient for this simulation. Once again, I am not convinced that this protein model ressembles the native protein, which makes any inference based on it questionable!

Lines 283-291: I think the authors misinterpreted the RoG figure. There is a lot of fluctuation in the "wild-type" protein, around 5A. This does not indicate that the wild-type protein was stable!

Line 296: Even with the flaws, the results are very poorly discussed.

Author Response

(i) Lines 62-64: This is a very delicate point, and the authors should be careful in order to state this. Rafaee et al. [26] for instance, stated that "However, the in silico based nsSNP predictions are not solely adequate for deriving genotype–phenotype relations and more experimental investigation is needed to elucidate the impact of the I661T changes on the AKAP3 structure in these patients"

Response: We thank the reviewer for pointing this out. We have corrected the narration that is: "Research studies [26–28] have applied bioinformatics tools in determining the association of demonstrated the efficacy of in silico techniques in associating diverse nsSNPs with diseases." (Pg 2, Line 64-65)

(ii) Lines 67-71: On what basis are the authors raising this hypothesis? I am not sure if this assumption makes any sense.

Response: We appreciate the reviewer for highlighting that point. We have provided missing references from the scientific literature that supports the contribution of CMAH and high level of Neu5Gc expression in promoting animal diseases that also jeopardize the health of humans. The references included are 1. doi.org/10.1371/journal.ppat.1007133, 2. doi.org/10.1111/xen.12260, and 3. doi.org/10.3390/v13050815. (Page 2, Line 67-70)

(iii)  Line 75: In general, the methodological procedures need to be better described, not just describe what each program does.

Response: The mentioned paragraph is presented below the introduction. We have corrected the paragraph narration and written the study objectives there. A complete methodology is described in the relevant section. (Page 2, Line 71-75)

(iv) Lines 99-102: The authors should specify what was examined in these tools. What kind of analyses were performed? What parameters were used?

Response: We have overall modified the methodology section. It is ensured that irrelevant information has been omitted, and the purpose of using the tools in the study and the method of performing the experiment using them are clearly stated. (Page 2-5, Line 84-245)

(v) Line 103: Due to the uncertain nature of the molecular modeling approaches used by the authors, it is critical that molecular dynamics simulations be performed to assess the stability of the predicted model and therefore determine which samples in the simulation are stable when compered to the initial model to be used in subsequent analyses. There is insufficient theoretical evidence that the protein model used in these analyzes is reliable.

Response: We appreciate the reviewer's comment and enhanced the simulation duration to 50ns from 20ns. MD simulations were performed through GROMACS 2016, and analysis was done by building trajectory. The analysis included RMSD, RMSF, the radius of gyrations, SASA, and hydrogen bond number. The overall analysis depicted that those mutations enhanced the stability of the CMAH. (Page 5, Line 224-246)

(vi) Lines 181-183: Is this categorization important? There is no mention of that in Introduction, Objetctive and Materials and Methods!?!

Response: The categorization describes the grouping of nsSNP into disease-causing and neutral/tolerant class that facilitates the prediction of pathogenic variants. The classification criterion is also explained as we have rewritten the materials and methods section. (Page 3-4, Line 142-150)

(vii) Line 184: How were the nsSNPs identified? This was not described in Materials and Methods!

Response:  We initially took DNA samples of 165 bovines from 29 breeds. We sequenced their DNA with Illumina HiSeq, mapped the mutations on UMD3.1/bosTau6, and aligned using Burrows-Wheeler Aligner (BWA) version 0.5.9-r16. We then re-mapped the mutations on ARS-UCD1.2 assemblies, resulting in the identification of novel variants and the confirmation of previously known variants through their corresponding RefSNP I.D. We have included the relevant methodology in the materials and methods section and incorporated UMD3.1 results in the supplementary file. (Page 2, Line 84-107).

(viii) Lines 194-201: This is not Results!

Response: We thank the reviewer for pointing out the text misplacement. Those sentences describe protein prediction methodology. We have omitted the mentioned sentences and have ensured that the protein prediction methodology is described in the materials and methods section.

(ix) Line 204: 60% of the protein was characterized as "others"! The authors should better characterize the structure of the protein!

Response: In response to the reviewer's concern regarding the classification of 60% of the protein as "others," we have taken additional steps to characterise the structure of the protein better. Firstly, we utilized the D2P2 tool to predict the presence of disordered regions within the protein structure (details in supplementary Figure S1). The analysis revealed a high degree of disorder, which could potentially account for a substantial portion of the "others" classification. Intrinsically disordered regions play crucial roles in various cellular processes, and their presence in the CMAH protein might contribute to its unique structural and functional properties. Moreover, we re-performed MD simulations to gain further insight into the protein's structural stability. In conjunction with the disorder prediction, these simulations allowed us to analyze the protein structure and address the reviewer's concern. (Page 14, Line 408-435)

(x) Figure 2: Although the assessment parameters used indicate satisfactory numbers, the protein 3D structure presents several long loops and turns arranged in an unconventionally conformation. Some CMAH proteins of other organisms have been modelled, and it is clear the difference, as one can see in PDB. Therefore, I am not sure if this model really ressembles the native protein.

Response: We thank the reviewer for thoroughly evaluating the manuscript. Before the study, we ensured the unavailability of bovine-CMAH protein structure in PDB. After predicting the structure, we attempted to superimpose it with available CMAH structures of other species in PDB, such as humans and mice. Unfortunately, CMAH has greatly evolved in species, so it has lost its function in most species. Due to lower homology, superimposition of the predicted structure was not possible. However, to address the reviewer's concern, we predicted the presence of disordered regions through the D2P2 tool showing a high disorder presence in this protein structure (Figure S1: Disordered region present on bCMAH protein predicted by D2P2). We also re-performed MD simulations that also facilitated in determining structure stability. (Page 14, Line 408-435)

(xi) Lines 273-282: The RMSD values are extraordinarily high. This indicates at least significant conformational changes if not at least partial unfolding, even for the "wild-type" model. In addition, the RMSD figure shows that 20 ns was not sufficient for this simulation. Once again, I am not convinced that this protein model ressembles the native protein, which makes any inference based on it questionable!

Response: We have re-performed the MD simulations and have also extended the MD simulation’s duration. Furthermore, we initially used forcefield Amber, which might have caused trouble in the simulation. We have used forcefield CHARMM36 to re-perform MD simulations by consulting the latest literature and MD tutorials. We have also incorporated new analyses such as RMSF, SASA, and number of hydrogen bonds, which further gave us insight into the impact of variations on the protein's structural stability. (Page 14, Line 408-435)

(xii) Lines 283-291: I think the authors misinterpreted the RoG figure. There is a lot of fluctuation in the "wild-type" protein, around 5A. This does not indicate that the wild-type protein was stable!

Response: We have conducted the MD simulations once more and extended the duration of these simulations to improve their accuracy. Previously, we employed the Amber forcefield, which may have introduced difficulties in the simulation process. We consulted current literature and MD tutorials to address this issue and used the CHARMM36 forcefield for our re-executed MD simulations. Additionally, we incorporated new analytical methods, such as RMSF, SASA, and the number of hydrogen bonds. These additional analyses allowed us to understand better how variations in the protein might impact its structural stability. (Page 14, Line 408-435)

(xiii) Line 296: Even with the flaws, the results are very poorly discussed.

Response: We have improved the discussion, so it is more targeted and has more clarity, as suggested by the reviewer. (Page 16-18, Line 439-554)

Round 2

Reviewer 3 Report

The authors have significantly improved the manuscript, addressing the previous reviewers' concerns. My only concern now is that I am not convinced by the results presented in the paper that all mutations enhanced the overall stability of the bCMAH protein, as concluded by the authors. MD simulations are showing only small deviations among wild-type and mutated proteins. Authors should review these results to draw appropriate conclusions as the data support. Also, the text needs to be proofread for minor formatting errors. Figure 6, for example, lacks information about the caption.

Author Response

The authors have significantly improved the manuscript, addressing the previous reviewers' concerns. My only concern now is that I am not convinced by the results presented in the paper that all mutations enhanced the overall stability of the bCMAH protein, as concluded by the authors. MD simulations are showing only small deviations among wild-type and mutated proteins. Authors should review these results to draw appropriate conclusions as the data support. Also, the text needs to be proofread for minor formatting errors. Figure 6, for example, lacks information about the caption.

Response:

Authors thank the reviewer for the positive and encouraging comment. We value your concern and that is why, we have incorporated additional analysis to determine the impact of mutations on CMAH protein stability. Webserver based tool, I-Mutant, was used that also showed the stabilising effect of all variations. We have also re-analysed our data, which shows that in comparison to wildtype, all mutations promoted stability, but mutation A210S has a more stabilising effect according to MDs. We have included the relevant methodology on page 4, lines 155-159, relevant results on page 12, lines 316-320 and page 12, lines 341-344, and the conclusion on page 15, 461-463. We have also corrected the Figure 6 caption and other minor formatting errors.